# Illicit Substance Use and the COVID-19 Pandemic in the United States: A Scoping Review and Characterization of Research Evidence in Unprecedented Times

**DOI:** 10.3390/ijerph19148883

**Published:** 2022-07-21

**Authors:** Anh Truc Vo, Thomas Patton, Amy Peacock, Sarah Larney, Annick Borquez

**Affiliations:** 1Johns Hopkins Bloomberg School of Public Health, Johns Hopkins University, Baltimore, MD 21205, USA; 2Division of Infectious Diseases and Global Public Health, University of California San Diego, La Jolla, San Diego, CA 92093, USA; tepatton@health.ucsd.edu (T.P.); aborquez@health.ucsd.edu (A.B.); 3National Drug & Alcohol Research Centre, University of New South Wales, Sydney, NSW 2052, Australia; amy.peacock@unsw.edu.au; 4Department of Family Medicine and Emergency Medicine, University of Montreal, Montreal, QC H3C 3J7, Canada; sarah.larney@umontreal.ca

**Keywords:** COVID-19, illicit drugs, substance use, scoping review, harm reduction

## Abstract

We carried out a scoping review to characterize the primary quantitative evidence addressing changes in key individual/structural determinants of substance use risks and health outcomes over the first two waves of the COVID-19 pandemic in the United States (US). We systematically queried the LitCovid database for US-only studies without date restrictions (up to 6 August 2021). We extracted quantitative data from articles addressing changes in: (a) illicit substance use frequency/contexts/behaviors, (b) illicit drug market dynamics, (c) access to treatment and harm reduction services, and (d) illicit substance use-related health outcomes/harms. The majority of 37 selected articles were conducted within metropolitan locations and leveraged historical timeseries medical records data. Limited available evidence supported changes in frequency/behaviors/contexts of substance use. Few studies point to increases in fentanyl and reductions in heroin availability. Policy-driven interventions to lower drug use treatment thresholds conferred increased access within localized settings but did not seem to significantly prevent broader disruptions nationwide. Substance use-related emergency medical services’ presentations and fatal overdose data showed a worsening situation. Improved study designs/data sources, backed by enhanced routine monitoring of illicit substance use trends, are needed to characterize substance use-related risks and inform effective responses during public health emergencies.

## 1. Introduction

In the United States (US), the confluence of the coronavirus disease 2019 (COVID-19) pandemic with the opioid overdose epidemic, which began over 20 years ago, has been appropriately termed a “clash of epidemics” [1]. US rates of both opioid and stimulant use are among the highest worldwide [2] and the COVID-19 mortality burden has been staggering compared to other high-income countries [3,4].

In the years leading up to the pandemic, the opioid crisis had been compounded by the growth in polydrug use and the illicit supply of fentanyl and other highly potent synthetic opioids [5,6]. Increasing rates of methamphetamine-related harms (most notably psychosis) represented another cause for concern [7,8]. The range of social and economic disruptions caused by the pandemic were anticipated to bring about further changes in substance use contexts among people who use drugs (PWUD) [9]. Policies restricting social movements directly influence substance use contexts, such as switching from communal to isolated substance use, whereas border shutdowns affect the flow in drug supply, which might modify the availability of different illicit substances [10]. Notably, the systemic shock to healthcare infrastructures imposed additional shifts in resource allocation and healthcare access for substance use treatment and harm reduction services [11]. The convergence of these multiple factors is expected to affect substance-use related health outcomes, including overdose, human immunodeficiency virus (HIV), and hepatitis C virus (HCV) incidence [12,13].

Whereas previous review articles have examined various interactions between the COVID-19 pandemic and substance use in the US [14,15,16,17], none have sought to systematically synthesize and appraise the quantitative evidence that emerged during the midst of this crisis on the impact of key changes in individual and structural level determinants of substance use risk and their associated health outcomes. A rigorous approach is therefore needed to describe it and support the formulation of unbiased inferences, particularly with regards to the priorities for future research and the implications for policymakers in responding to similar crises in the future within US communities. We undertook a scoping review, including systematic data extraction and visualization of quantitative data capturing changes in illicit substance use following the onset of the COVID-19 pandemic and up to the end of the second wave (based on deaths counts) in August 2021 [18], also corresponding to when over 50% of the population had been fully vaccinated and to the start of the booster schedule in the US [19]. Our focus on this time frame during the pandemic was to emphasize the evidence body before the adaptation of individuals, services, and the research process to the pandemic environment, wherein more abrupt and drastic changes relating to substance use and related outcomes would be expected, as well as during which the research infrastructure would be less prepared to crisis conditions. Our scope for a US-specific review was motivated by the need to frame findings within a specific COVID-19 pandemic, substance use, healthcare, surveillance, and research environment that would allow for comparisons and recommendations to be made. We set on four key questions based on previous research [20]: during the midst of the COVID-19 pandemic, (a) did illicit substance use frequency, contexts and behaviors change?, (b) did illicit drug market dynamics change?, (c) did access to substance use-related healthcare and harm reduction services change?, and (d) did substance use-related health outcomes/harms change?

## 2. Methods

Based on guidance from Munn et al. and Peters [21,22], we set out to provide a scoping review of the peer-reviewed literature that focuses on the consequences of COVID-19 and illicit substance use for individuals in the US and applied the PCC (Population, Concept, and Context) framework to guide our question development (Appendix B). As an increasingly more popular research synthesis tool, a scoping review is designed for the generalized mapping of key concepts and snapshot of the available evidence while helping to shed further insights on the landscape of multifaceted, evolving issues. The format of the scoping review allows for both a systematically guided and structured methodology to review evidence that could clarify broader topics, which aligns with the goal of our study. Detailed PRISMA-ScR process guidelines [23], as followed in this review, are provided in Appendix D Table A1. We opted to employ LitCovid, an open literature hub sourced from PubMed that curates COVID-19-specific published articles or studies and is maintained through daily updates indexing from the larger PubMed database [24].

We developed a search strategy with the broadest substance use-related key terms (Appendix C) without date restrictions. The latest update to the search was on 6 August 2021. During the screening stages, articles were considered ineligible if they: (1) did not mention the relationship between illicit substance use and COVID-19 or the pandemic consequences, and (2) did not present evidence relevant to at least one of the four questions posed in the previous section. “Illicit substances” were defined as being inclusive of all classes of controlled drugs that are not available for purchase in legal retail markets in the US. Hence, articles presenting results focusing on tobacco, alcohol, or cannabis/marijuana alone were considered to be beyond the scope of this review as they are available for legal purchase in all or some US states. Articles were also excluded if they were qualitative studies, were individual case reports, were feasibility/pilot studies, did not provide primary data or took place outside of the US. Full protocol details for the double screening process carried out by AV and TP can be found in Appendix C.

Relevant study descriptions and main findings were systematically extracted (Appendix A). No formal risk of bias assessment was conducted but study design was presented in place [21]. Where measures were deemed to be sufficiently similar across studies (proportions or statistical associations quantifying a specific outcome), results were visualized in the form of infographics. These figures served not as results of meta-analyses, but simply as visual summaries of the data extracted (often following calculation) from available studies to enable more straightforward interpretations of available evidence.

## 3. Results

Our search strategy returned a total of 2440 articles. After the title/abstract screening and full-text review processes, 37 articles were included based on our criteria (Figure 1).

As detailed in Table 1, the most common geographical research setting was the national level (32%) followed by the Northeast (27%). In addition, most studies employed a retrospective pre-post design (67%) and a large share (35%) utilized a range of medical records.

Figure 2 shows the number of studies published over time stratified by the type of study design. Table 2 summarizes the characteristics of the studies and Table 3 provides relevant data for each of the four questions across all studies included.

### 3.1. Changes in Illicit Substance Use Frequency, Behaviors, and Contexts

A fifth (n = 9) of the reviewed studies provided evidence pertaining to changes in illicit substance use frequency, contexts, and associated risk behaviors. For this domain, we make a distinction between studies that explicitly sampled people with a history of illicit substance use or substance use disorders (SUD) and those that sampled people without any prerequisites regarding their history of substance use.

Studies that reported on similar quantitative outcomes are presented in Figure 3 and Figure 4, with study information provided in Table 2 and Table 3 and detailed outcomes outlined in Appendix A.

#### 3.1.1. Changes in Illicit Substance Use Frequency

Two cross-sectional studies recruited participants with an established history of substance use or SUD. Jacka et al. [25] looked at patients, surveyed between May and June of 2020, receiving medications for opioid use disorder (MOUD) and found that the proportions of patients reporting increased (38%) and decreased (42%) illicit substance use during the COVID-19 pandemic were comparable. The proportion of patients reporting an increase in substance use was shown to be much higher among those with higher SUD severity [25]. Mistler et al. [26] compared the impacts of the pandemic on changes in substance use frequency in a sample of patients recruited from a methadone clinic between May and October, 2020. Although most survey respondents reported no change in their use of non-prescription drugs, a higher proportion of respondents in the racial-ethnic minority group (23%) reported a decrease in non-prescription drug use compared to their White counterparts (4.5%) [26].

Among populations for whom recruitment was not related to history of substance use or SUD, we observed mixed trends in illicit substance use during the pandemic. Janulis et al. [27] presented findings from a longitudinal cohort study following young men who have sex with men (MSM) and young transgender women in Chicago and showed that the prevalence of illicit substance use declined during the pandemic, based on survey responses between 21 March and 1 October 2020, compared to survey responses provided at any point in time in the 78 weeks leading up to 21 March 2020 (OR = 0.61, 95% CrI 0.37–0.96). A study by Starks et al. [28] presented findings from an online survey conducted between 6–17 May 2020 in a sample of sexual minority men and found significant decreases in the prevalence of illicit substance use during the pandemic compared to a matched sample of respondents surveyed pre-COVID. The prevalence of methamphetamine, methylenedioxy-methamphetamine (MDMA), gamma hydroxybutyrate (GHB), ketamine, and cocaine/crack use all showed decreases. Cocaine use prevalence was also found to be lower during the pandemic, based on a survey of people recruited from the Miami Adult Studies on HIV from 32.7% between July and August of 2020, compared to 14.3% inn survey responses provided 7.3 ± 1.5 months earlier [29]. The fall in cocaine use was shown to be greater in a subgroup of participants living without HIV (41.2% to 15.0%) compared to participants living with HIV (26.7% to 13.8%) [29].

Palamar and Acosta [30] reported results from a cross-sectional survey of electronic dance music adult partygoers who were recruited between 18 April and 25 May 2020. This study found that most participants reported a decreased frequency of cocaine use (78.6%), ecstasy/MDMA/Molly use (71.1%), and lysergic acid diethylamide (LSD) use (68.0%) following the onset of the COVID-19 crisis, which was assumed to have started during the week of 16 March 2020, with the remainder in each case reporting either an increase or no change.

In contrast, a study by Duncan et al. [31] compared self-reported substance use among individuals discharged from Hennepin County Jail in Minnesota during the pandemic (April and May 2020) to equivalent data collected prior to the pandemic (January and February 2020) and found the proportion self-reporting the use of cocaine, methamphetamine, and heroin all increased following the onset of the pandemic.

Two studies provided insight from toxicology data. Young et al. [32] described relative changes in urine toxicology results from traumatically injured patients just before (1 January 2020–18 March 2020), during (19 March 2020–30 June 2020), and a year prior to the pandemic (19 March 2019–30 June 2019) in Southern California. Positive toxicology rates for any type of drug were higher during the pandemic compared to the pre-pandemic periods, as were those specifically relating to amphetamines and MDMA [32]. When comparing positive toxicology rates during the pandemic to those from the historical control period, there were decreases for opioids and increases for cocaine [32]. In addition, Niles et al.’s study [33] comparing drug test results collected before (1 January 2019–14 March 2020) and after (14 March 2020–16 May 2020) the onset of the pandemic among a national sample of US adults in Quest Diagnostics database found the volume of weekly drug testing declined by 70% after stay-at-home orders were introduced. They found significant increases in positive test rates for fentanyl, heroin, and opiates following the onset of the pandemic but neither change nor a reduction in positive tests for drugs such as amphetamines, oxycodone, and benzodiazepines [33].
Figure 3Summary evidence for changes in illicit substance use frequency associated with the pandemic. PCP = Phencyclidine. LSD = Lysergic Acid Diethylamide. MDMA = Methylenedioxy-Methamphetamine. GHB = Gamma Hydroxybutyrate. PLHIV = People Living with HIV. MSM = Men Who Have Sex with Men. CT = Connecticut. MN = Minnesota. IL = Illinois [26,27,28,29,31,32,33].
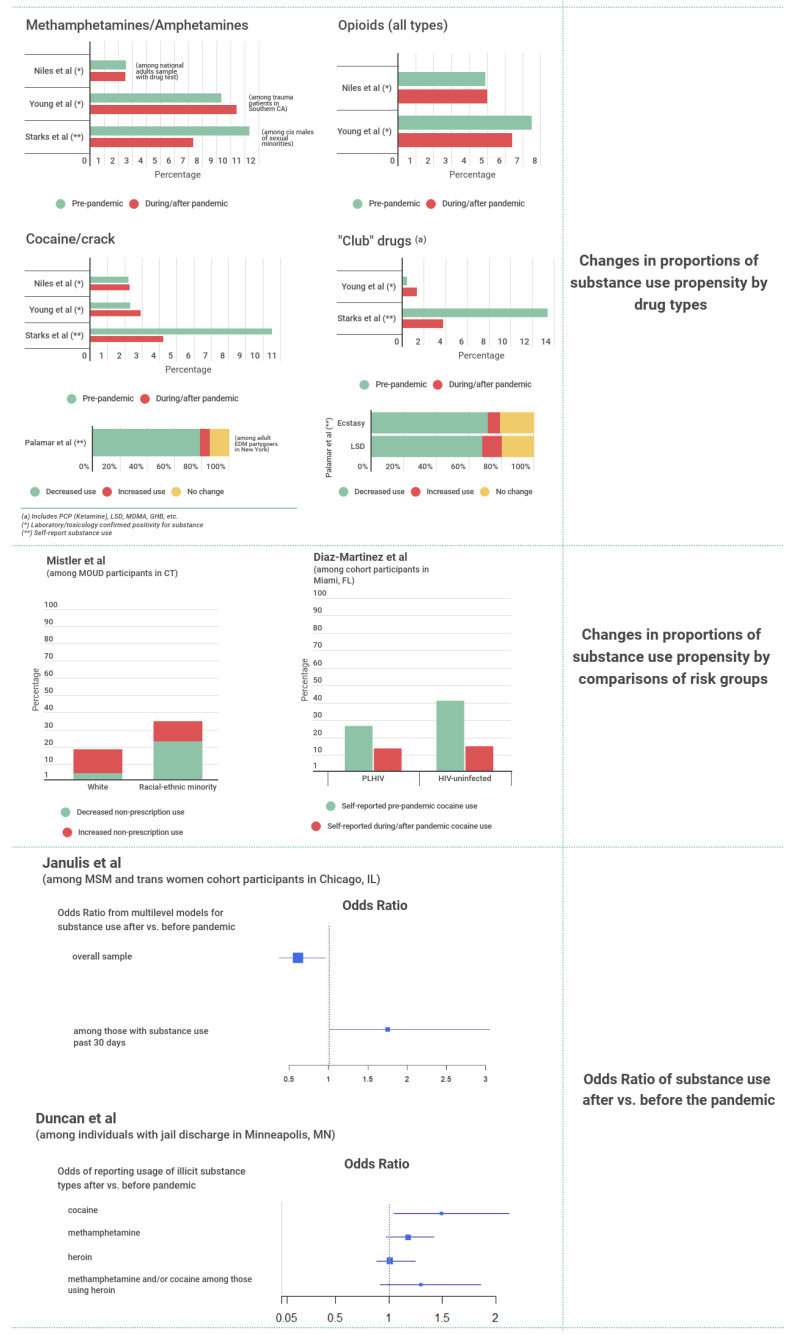


#### 3.1.2. Changes in Illicit Substance Use Contexts and Related Behaviors

Niles et al. [33] also found evidence of increased use of dangerous substance combinations during the pandemic, (Figure 4) as the proportion of positive tests for non-prescribed fentanyl alone and in combination with amphetamines, benzodiazepines, cocaine, opiates, and heroin increased significantly across all demographic groups during COVID-19 compared to the months before the stay-at-home orders.

There is scant evidence regarding changes in interpersonal aspects or contexts of substance use. Mistler et al. [26] found a majority of all participants in the study reporting no change in the sharing of drugs or equipment, with 12.7% reporting a decrease in drugs or equipment sharing and none reporting an increase in such behavior. A majority of participants also reported no changes in condomless sex or in seeking transactional sex [26].

Among sexual minority men, Starks et al. [28] showed that the association between substance use and both the number of casual sexual partners and condomless anal sex was significantly greater during the pandemic compared to before (*p* < 0.01).
Figure 4Summary evidence for changes in illicit substance use contexts and behaviors. MOUD= Medications for Opioid Use Disorders. CT= Connecticut [26,33].
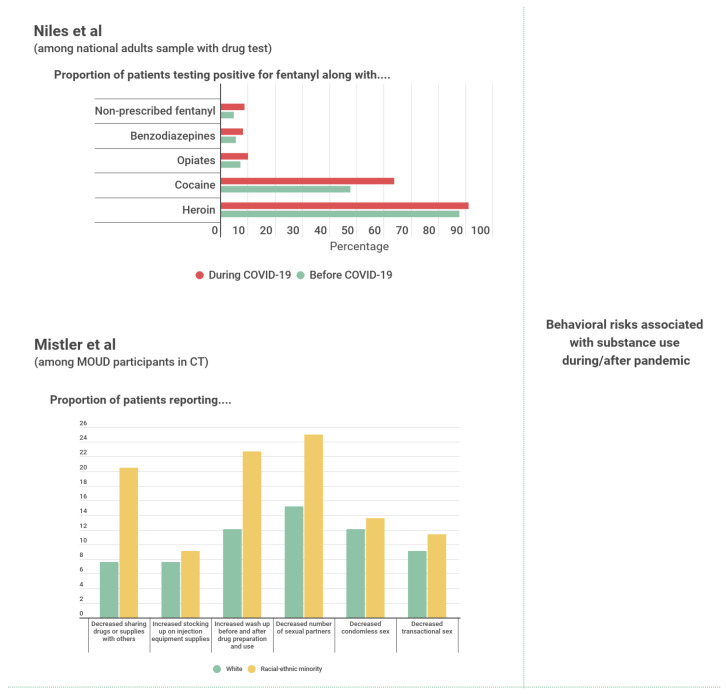


### 3.2. Changes in Illicit Drug Market Dynamics

Only four articles addressed changes in illicit drug market dynamics, with available quantitative results detailed in Appendix A, as well as selected information provided in Table 2 and Table 3, and summary graphic evidence presented in Figure 5.

Palamar and Acosta’ [30] survey of partygoers provided varied evidence regarding substance cost and quality as reported by participants. The study found that more people reported an increase, as opposed to a decrease, in the cost of cocaine and ecstasy, coupled with a decrease in the quality of their preferred substance, although the majority indicated there being no change [30]. Similarly, 9.7% of participants from the Miami Adult Studies of HIV [29] reported an increase in the price of cocaine, whereas 7.2% reported greater difficulty accessing cocaine during the past month. Jacka et al. [25] recorded a 0.19 overall probability of participants reporting difficulty accessing their preferred substances in their study sample.

Another study by Palamar et al. [34] presented evidence on drug seizures for five High Intensity Drug Trafficking Areas (HIDTA) in the US. The authors found that the monthly number of law enforcement-related drug seizures, particularly for methamphetamine, decreased during the 12 months pre-pandemic, and then increased significantly from March 2020 to September 2020 [34]. The study found that the monthly number (and weights) of fentanyl seizures increased during the months before the pandemic and followed the same trend during the pandemic. In contrast, there were steady decreases through the pandemic in the weight of heroin seizures [34].
Figure 5Summary evidence for illicit drug market changes. MOUD= Medications for Opioid Use Disorders. LSD = Lysergic Acid Diethylamide. EDM = Electronic Dance Music. NY = New York [25,34].
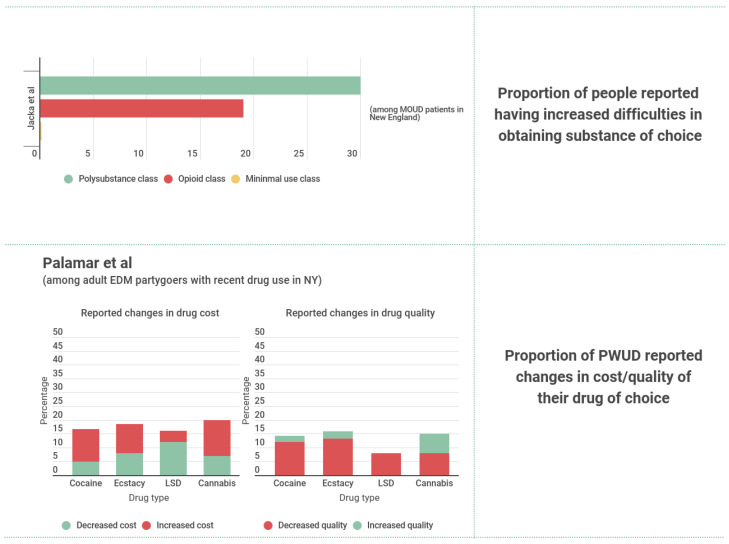


### 3.3. Changes in Illicit Substance Use-Related Treatment and Harm Reduction Services Access

With 16 out of 37 articles providing data related to changes in treatment and harm reduction services access, the majority utilized patients’ medical records retrospectively, either from MOUD clinics or state and national prescription databases, to ascertain desired outcomes.

#### 3.3.1. MOUD Treatment Services

In recognition of pandemic conditions, the Substance Abuse and Mental Health Services Administration (SAMHSA) issued regulatory changes in March 2020 around the delivery of MOUD, including increases in the number of take-home doses allowed as well as lifting administrative limits placed on the number of new patients for whom providers can issue prescriptions [35]. The effect of these policy changes can be seen in Amram et al.’s [36] findings of increased average take-home dose per client during the pandemic in Spokane, WA, and Jones et al.’s [37] finding that a third of clinicians from an email survey reported prescribing to new patients during the pandemic without in-person examination. Caton et al. [38] also found that 65% of the 57 MOUD primary care clinics in their study reported longer buprenorphine prescribing periods, whereas Brothers et al. [39] found increases between 19% to 16,700% in the proportion of Connecticut patients receiving 3 to 28-day take home dose supplies during the pandemic compared to before. Finally, Hughes et al.’s [40] data from office-based opioid treatment services at a clinic in the Appalachian region showed increases in the total number MOUD visits following the pandemic (*p* < 0.001). Contrastingly, other studies also indicate reduced access or capacities for MOUD programs in some settings despite the regulatory changes. This includes Herring et al.’s [41] report of a 34% decline in referrals to a buprenorphine treatment program across 52 California hospitals as well as a 48% decrease in the number of patients prescribed buprenorphine after the initiation of stay-at-home orders. Bandara et al.’s cross-sectional study [42] surveyed 16 carceral systems and found that 10 were faced with a reduction in MOUD program scale during the pandemic. In addition, Downs et al.’s [43] reported a significant decrease in both the number of patients who filled new opioid or benzodiazepine prescriptions in Texas. Lastly, Nguyen et al.’s [44] report from analyses of a nationwide retail pharmacy database found a significant, 0.50% decrease in the weekly growth rate in buprenorphine prescription fills following the pandemic (95% CI −0.84% to −0.15%).

The review also found several studies reporting the degree of success in the continuation of treatment services by healthcare providers for people with SUD. Among a sample of individuals with jail discharges, Duncan et al. [31] found an increase in the odds of receiving buprenorphine (initiation, maintenance, or taper) during their incarceration period after the COVID-19 pandemic (Figure 6). Caton et al. [38] also found that a third of clinics in their survey reported easier treatment retention during the pandemic, with 17.5% of the clinics also reporting increased demand for medication visits. In addition to the general decrease in buprenorphine referrals, Herring et al. [41] found a 33% decrease in the number of patients attending a follow-up visit for SUD treatments after the implementation of stay-at-home orders in California.

#### 3.3.2. Other Harm Reduction Services

Evidence on changes in the delivery of harm reduction services following the onset of the pandemic was limited, although multiple studies indicated that there were disruptions in access to these services. Jones et al. [45] analysed medical records data from the IQVIA Total Patient Tracker database for all patients prescribed buprenorphine products between March to May 2020. Their study found that the number of unique patients dispensed ER intramuscular naltrexone was significantly lower than forecasted estimates, ranging from 1039 to 2498 fewer patients in March and May, respectively. Jacka et al. [25] highlighted difficulties in accessing naloxone and needle exchange among 7–8% of all participants in their study in New England. Furthermore, Glenn et al. [46] found that patients who received naloxone via emergency medical services (EMS) in Tucson, Arizona were more likely to refuse transportation during the pandemic period (16 March 2020–30 April 2020) compared to pre-pandemic (1 January 2020–15 February 2020). However, Zubiago et al.’s [47] study looking at hospitalized PWUD at Tufts Medical Center in Boston, Massachusetts, indicated greater odds of receiving an HIV test among patients with stimulant use (OR 2.7, 95% CI 1.9 to 3.81) and benzodiazepine use (OR 1.53, 95 % CI 1.03 to 2.23) versus pre-pandemic.

### 3.4. Changes in Substance Use-Related Health Outcomes

With a total of 15 studies identified with outcomes related to this domain, most evidence addressing this question reported outcomes related to fatal or non-fatal overdose from opioids.

#### 3.4.1. Emergency Services Utilization for Illicit Substance Use and Non-Fatal Opioid/Other Substance-Related Overdose Outcomes

Five studies presented findings on non-fatal overdose outcomes from data at the national level. Holland et al. [48] presented evidence from CDC surveillance data showing that the mean ED visit rate associated with opioid overdoses was significantly higher during the pandemic (15 March 2020–10 October 2020) compared to the equivalent period in 2019. An analysis of the comprehensive national EMS registry (NEMSIS) by Handberry et al. [49] revealed that the percentage of all EMS activations related to opioid use increased from 0.6% to 1.1% (a total increase of 1538 activations) during the first 10 weeks of the pandemic (*p* < 0.001), then gradually decreased to 0.7% by the last week of December 2020.

Data from 18 emergency departments (ED) in 18 different US states [50] would appear to contradict the studies described previously, as the 2020/2019 ratio of opioid-related ED visits, for equivalent time periods between January-June, was 0.82. For ED visits related to other SUD, the ratio was 0.84 [50]. However, these results do not account for the overall decreasing trend in ED visits. The same data can be presented to show that the percentage of all ED visits related to opioid use and to other drugs increased from 0.5% to 0.6% and from 1.0% to 1.4%, respectively. Soares et al. [51] examined changes in ED visits following a non-fatal opioid overdose during the pandemic (1 January 2020–12 December 2020) compared to historical controls (1 January 2018–12 December 2019) in Alabama, Colorado, Connecticut, North Carolina, Massachusetts, and Rhode Island. The results show a significant increase of 28.5% (95% CI 23.3–34.0) in the rate of ED visits for opioid overdose per 100 all-cause ED visits in 2020 compared to the data from the period between 2018–2019 [51]. Lucero et al. [52] analyzed ED records from 16 US states and found a 9.3% decrease in the volume of substance use-related ED encounters after the shelter-in-place orders were in effect. Despite the authors attributing this underutilization phenomenon to hesitancy to engage in the healthcare system during the pandemic, we note that such a drop was also the slightest compared to other types of ED encounters (39.6% decrease in overall ED volume) [52].

An additional five studies presented findings on non-fatal overdose outcomes using region- or facility-specific data (i.e., not at the national level). Following national trends, in Louisville, Kentucky, Shreffler et al. [53] showed that despite total ED visits having decreased by 21.3% during the pandemic (March 2020 to June 2020), the average weekly ED overdose count was 24% higher than that for the period immediately prior to the pandemic (November 2019 to March 2020) and 31% higher than that in the equivalent period in 2019 (March 2019 to June 2019). Evidence from ED data in Vermont [54] showed that the mean number of ED visits for overdose increased slightly between the pre-pandemic and peri-pandemic periods (Difference = 2.6, 95% CI −4.8–10.1). Meanwhile, EMS data from Guilford County, North Carolina [55] shows that there was a significant increase (37.4%) in mean weekly opioid overdose-related EMS runs during the pandemic period compared to before, as well as a significant increase (24.3%) compared to the mean number of runs in the same 29 weeks of previous year as reference. In contrast, a study by Rosenbaum et al. [56] of non-fatal opioid overdoses in an urban healthcare system in Philadelphia, Pennsylvania, found a 6.8% decline during the first 100 days of a shelter-in-place order compared to the 100 day period preceding it (*p* < 0.001). However, as with the study by Pines et al. [50], the same data can be interpreted differently to show that the percentage of all ED visits attributable to opioid overdoses increased from 3.6% to 4.3% [56]. Data from 129,429 ED encounters within 3 southwestern Connecticut hospitals [57] also found that mental health-related ED encounters involving opioid withdrawals was lower in 2020 compared to 2019 in absolute terms. Once again, the same data can be reinterpreted such that the percentage of all ED encounters attributable to opioid withdrawals increased from 4.6% to 5.4% [57]. The same study showed that encounters involving other psychoactive substance use increased in 2020 compared to 2019 [57]. Relative percentage changes of overdose and substance use-related ED/EMS numbers can be found in Figure 7.

One study, Ridout et al. [58], provided data for non-alcohol related SUD psychiatric visits in lieu of overdose-related outcomes, and found a 47.8% (95% CI 42.7% to 52.8%) higher number of visits in 2019 compared to post-pandemic period among outpatient psychiatric care seekers in a large Northern California hospital system.

#### 3.4.2. Fatal Opioid/Other Substance-Related Overdose Outcomes

Five studies presented findings on fatal substance-related overdoses. Shreffler et al. [53] analyzed data from Jefferson County Coroner’s Office (Kentucky) and showed that the average weekly overdose death counts were 45% higher during the pandemic (March 2020 to June 2020) compared to the period immediately prior (November 2019 to March 2020) and 88% higher than the equivalent period for the previous year (March 2019 to June 2019). Grunvald et al. [54] also found a significant increase in the mean number of monthly overdose-related fatalities during the pandemic (March 2020–May 2020) at the University of Vermont Medical Center compared to the pre-pandemic period (February 2019–February 2020). An assessment of vital records from the Ohio Department of Health in Vieson et al. [59] showed that there was a significantly higher opioid overdose death rate during the second quarter of 2020 (14.2) compared to the peak period in 2017 (8.3), as well as the rate of the first quarter of 2020. Data from the Medical Examiner’s Office in Cook County, Illinois, analyzed by Mason et al. [60], showed that the mean number of opioid overdose deaths per week peaked during the first three months of a stay-at-home order, then fell to pre-pandemic levels during the latter half of 2020. Although this trend was also observed for deaths involving heroin, fentanyl-related deaths continued to rise throughout 2020 [60]. In contrast, opioid-related fatalities and methadone-related fatalities decreased, albeit non-significantly, following the onset of the pandemic in Connecticut [39] (Figure 7).
Figure 7Summary evidence for substance use-related health outcomes. ED= Emergency Department. EMS= Emergency Medical Services. NC = North Carolina. PA = Philadelphia. KT = Kentucky. OH = Ohio. CT = Connecticut. VT = Vermont [39,48,51,53,54,55,56,59,60].
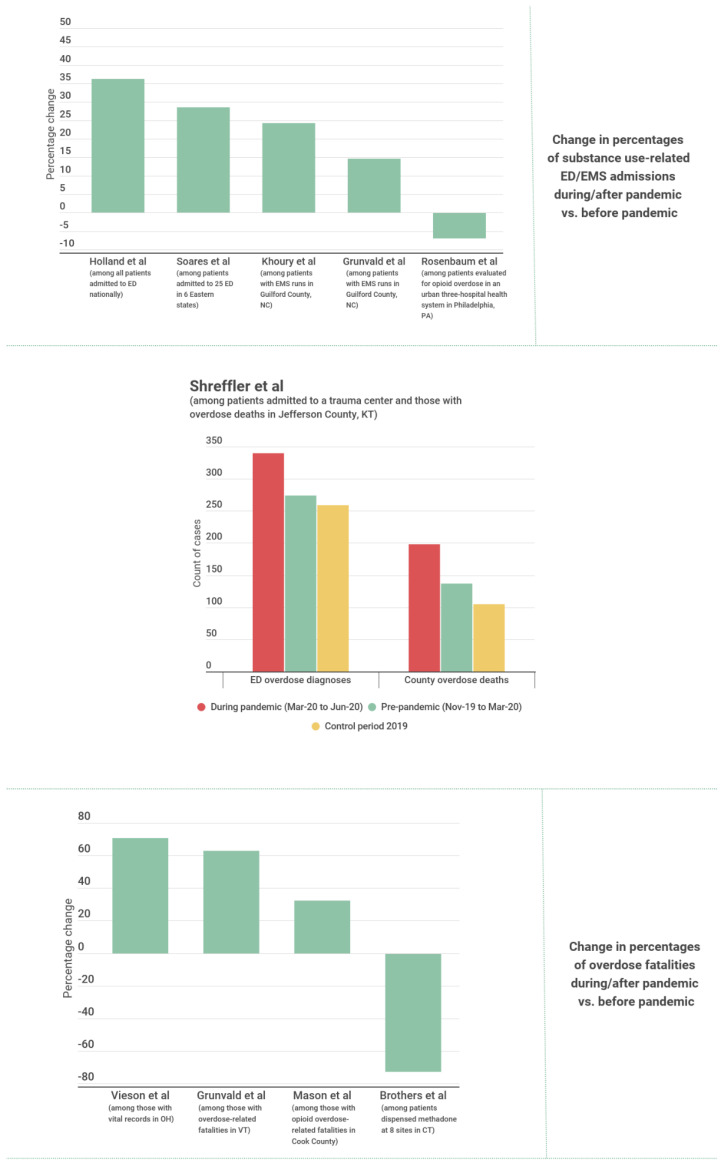

ijerph-19-08883-t002_Table 2Table 2Study Characteristics.Author (Year)Study DesignData SourceStudy PopulationSample SizeRegionData Pre and Post Pandemic Time Period of Data CollectionAmram et al. (2021) [36]Retrospective observationalPre-post convenience survey; Medical records—patient informationEnglish-speaking patients 18+ dispensed methadone at a Spokane opioid treatment program (OTP)249 patientsSpokane County, WashingtonYes1 December 2019–29 February 2020 (Pre-pandemic); 1 April 2020–30 June 2020 (Post-pandemic)Bandara et al. (2020) [42]Cross-sectionalEmail surveyLeaderships from 16 carceral systems identified as potentially initiating OAT16 eligible systemsUSNo5 May 2020–20 May 2020Brothers et al. (2020) [39]Retrospective censusComprehensive state-wide survey of Connecticut opioid treatment programs; State-level autopsies/toxicology reports on confirmed opioid-involved deathsPatients dispensed methadone at opioid treatment programs (OTPs); people at risk for opioid-involved deaths in study region24,261 patients served at 8 OTPs in Connecticut; All confirmed opioid involved deaths in ConnecticutConnecticutYes1 January 2015–31 December 2019 (Pre-pandemic); 1 January 2020–31 August 2020 (Post-pandemic)Buchheit et al. (2021) [61]LongitudinalClinic dataPatients receiving low threshold SUD treatment services at a clinicNAPortland, OregonYesJanuary 2020–August 2020Caton et al. (2021) [38]Cross-sectionalOnline surveyPrimary care clinics enrolled in an existing medication for opioid use disorder (MOUD) treatment expansion project57 clinicsUSNo20 April 2020–8 May 2020Diaz-Martinez et al. (2021) [29] LongitudinalPhone survey with sample based on recruitment to ongoing studyPeople living with and without HIVWhole sample = 196Miami, FloridaYesDuring participant’s last baseline visit (Pre-pandemic); July and August of 2020 (Post-pandemic)People living with HIV = 116HIV-uninfected people = 80Downs et al. (2021) [43]Retrospective observationalTexas PMP registry count of number of patients filled prescriptions of either an opioid or benzodiazepine productPatients filled prescriptions of either an opioid or benzodiazepine product in study regionAll unique patients filling new opioid prescriptions each dayTexasYes5 January 2020–20 March 2020 (Pre-pandemic, before restriction order on elective medical procedure); 31 March 2020–12 May 2020 (Post-pandemic, after restriction order)Duncan et al. (2020) [31]Retrospective observationalCensus data from Hennepin County Jail medications for opioid use disorder programIndividuals with jail discharge accounted at the Hennepin County Jail in study region4912 discharges (Pre-pandemic); 2794 discharges (Post-pandemic)Minneapolis, MinnesotaYes1 January 2020–29 February 2020 (pre-pandemic); 1 April 2020–31 May 2020 (Post-pandemic)French et al. (2021) [62]Cross-sectionalOnline surveyParticipants requested and received naloxone medication from a free mailed program in study region422 survey respondentsPhiladelphia, PANo1 March 2020–31 January 2021Glenn et al. (2021) [46]Retrospective observationalEMS system dataPatients receiving naloxone by EMSSample size pre-covid: 164Tucson, ArizonaYesPre-pandemic: 01 January 2020 to 15 February 2020Sample size during-covid: 153During-pandemic: 16 March 2020 to 30 April 2020Grunvald et al. (2021) [54]Retrospective observationalMedical records—ED; State and Chittenden County overdose reportsPatients 18+ presenting to ED for signs of opioid use disorder who meet inclusion criteria under the ED program that initiate medication for opioid use disorder (STAR) at University of Vermont Medical Center; Vermont residents with overdose-related fatalities 126 (Pre-pandemic); 4 (During/Post-pandemic); All overdose deaths in reportVermontYes1 February 2019–29 February 2020 (Pre-pandemic); 1 March 2020–31 May 2020 (Post-pandemic)Handberry et al. (2021) [49]Retrospective observationalMedical records—EMSPatients included in the NEMSIS database 16M (2018); 22M (2019); 25M (2020) total activationsStates and territories participated in reporting to NEMSIS, USYes1 January 2018–29 February 2020 (Pre-pandemic); 1 March 2020–31 December 2020 (Post-pandemic)Herring et al. (2020) [41]Retrospective pre-postData from CA BridgePatients identified with and treated for OUD monitored by the California Bridge initiative across a subset of 52 hospitals 70 participating hospital inpatient units across study regionCaliforniaYes30 December 2018–16 March 2020 (Pre-pandemic); 17 March 2020–10 October 2020 (Post-pandemic) Holland et al. (2021) [48]Retrospective pre-postCDC’s National Syndrome Surveillance Program data Patients presenting at ED in study region187,508,065 total ED visitsUSYes1 May 2019–31 March 2020 (Pre-pandemic); 1 April 2020–31 April 2020 (Post-pandemic)Hughes et al. (2021) [40]Retrospective observationalMedical records—patient informationPatients who had ever been prescribed a buprenorphine-containing medication and had an ICD-10 diagnosis code for OUD in EHR system at a single-family medicine clinic with a high concentration of providers that offer office-based opioid treatment (OBOT) services in a primarily rural and micropolitan region with a high overdose rate in study region242 patientsAppalachian MountainsYes16 January 2020–15 March 2020 (Pre-pandemic); 16 March 2020–15 April 2020 (Transition); 16 April 2020–15 June 2020 (Post-pandemic)Jacka et al. (2021) [25]Cross-sectionalSurveyPatients in 8 opioid treatment programs part of a hybrid trial Project MIMIC, who were 18+ and newly inducted on MOUD within the past 30 days135 respondentsNew EnglandNoMay 2020–July 2020 Janulis et al. (2021) [27]Longitudinal cohortCohort study dataYoung men who have sex with men and young transgender women part of the study cohort458 study participantsChicago, IllinoisYes21 March 2020–1 October 2020Jones et al. (a) (2021) [37]Cross-sectionalEmail surveyDATA-waived physicians identified through DEA files 10,238 cliniciansUSNo23 June 2020–19 August 2020Jones et al. (b) (2021) [45]Retrospective observational; Comparative Medical records—patient informationPatients dispensed OUD captured in the IQVIA Total Patient Tracker databaseAll patients dispensed buprenorphine products in data source during the study period (national sample)USYes1 January 2019–31 May 2020Khoury et al. (2021) [55]Retrospective observationalMedical records—EMSPatients with incident included in the study region’s EMS databaseAll occurrences of opioid-related EMS runsGuilford County, North Carolina Yes1 September 2014–9 March 2020 (Pre-pandemic); 10 March 2020–30 September 2020 (Post-pandemic)Lucero et al. (2020) [52]Retrospective, observational, cross-sectionalBilling dataIndividuals involved in emergency department encountersPre- shelter-in-place (SIP) order ED encounters = 25,884,38416 states within the USYes1 January 2017–20 April 2020Post-SIP order ED encounters = 339,054Mason et al. (2021) [60]LongitudinalCook County Medical Examiner’s Office dataOpioid-Involved Overdose FatalitiesA total of 4283 opioid overdose fatalities occurred during study periodCook County, IllinoisYesFour time periods:(1) 5 January 2018–3 December 2019;(2) 4 December 2019–20 March 2020;(3) 21 March -5 June 2020;(4) 6 June -23 December 2020Mistler et al. (2021) [26]Cross-sectionalPhone surveyParticipants from parent study recruited from Connecticut’s largest addiction treatment setting providing MOUD110 patientsConnecticutNo7 May 2020–18 September 2020Nguyen et al. (2020) [44]Retrospective observationalRetail pharmacy claims databaseIndividuals who filled prescriptions92% of retail pharmacy claimsUSYesEvery week between 1 May 2019, and 28 June 2020 except for the week of 8 March to 15 March 2020, which was excluded because this was the week before the transitioning week (16 March).Niles et al. (2021) [33]Retrospective observationalNational clinical laboratory databasePresumptive immunoassay screening tests872,762 specimensAllYesBaseline time period: 1 January 2019–14 March 202050 states and the District of ColumbiaCOVID-19 pandemic time period: 15 March–16 May 2020Palamar and Acosta (2021) [30]Cross-sectionalOnline surveyElectronic dance music (EDM) adult partygoers who live in study region and reported recent drug use128 participantsNew YorkNo18 April 2020–25 May 2020Palamar et al. (2021) [34]Retrospective pre-postHigh Intensity Drug Trafficking Areas drug seizure dataDrug seizure (cocaine, meth, heroin, fentanyl) accounts in study regionsAll drug seizuresWashington DC/Baltimore, Chicago, Ohio, New Mexico, and North FloridaYes1 March 2019–30 September 2020Pines et al. (2021) [50]Retrospective observationalData from 18 general U.S. acute care hospital EDsED visits for substance use disorders4.5 million ED visits18 US. statesYesJanuary–July 2019, January–July 2020Ridout et al. (2021) [58]Retrospective observationalMedical records—EHRPatients seeking outpatient psychiatric care at Kaiser Permanente Northern CA94,720 (2019); 94,589 (2020)Northern CaliforniaYes9 March 2019–31 March 2019 (Pre-pandemic); 9 March 2020–31 March 2020 (Post-pandemic)Rosenbaum et al. (2021) [56]Retrospective observationalMedical records—EHRPatients seen and evaluated for opioid overdose in an urban three-hospital health system in study region46,078 (Pre-pandemic); 35,971 (Post-pandemic)Philadelphia, PAYes14 December 2019–22 March 2020 (Pre-pandemic); 23 March 2020–30 June 2020Shreffler et al. (2021) [53]Retrospective observationalElectronic medical health record and county coroner dataPatients presenting to trauma center with an overdose diagnosis873 individuals had an overdose diagnosis in the ED and 440 individuals in the county diedof drug overdoseTrauma center at University of Louisville in Jefferson County, KentuckyYes16 weeks from the date that a state of emergency was declared by the governor (6 March 2020). This is Period 3. Compared with:- Same time period in 2019 (Period 1)- 16 weeks prior to 6 March 2020 (Period 2)Soares et al. (2021) [51]Retrospective observationalMedical records—EDAll adult ED visits to one of the 25 EDs across 6 health systems during study periods1,215,250 visits (2018); 1,283,303 visits (2019); 1,074,936 visits (2020)Connecticut, North Carolina, Colorado, Massachusetts, Alabama, Rhode IslandYes1 January 2018–31 December 2019 (Pre-pandemic); 1 January 2020–31 December 2020 (Post-pandemic)Starks et al. (2020) [28]Cross-sectional; ComparativeOnline survey18+ cisgender sexual minority males who indicated their partner was cisgender male, excluding those reporting vaginal sex455 (Pre-pandemic); 455 (Post-pandemic)USYes1 November 2017–30 November 2019 (Pre-pandemic); 6 May 2020–17 May 2020 (post-pandemic)Stroever et al. (2021) [57]Retrospective observationalMaster patient index data, a combination of clinical, financial, and administrative recordsED encounters attributed to mental health conditions in individuals 18 years and older seeking medical care129,429 ED encounters3 southwestern Connecticut hospitalsYesJanuary–August 2019 and January–August 2020Vieson et al. (2021) [59]Retrospective observationalVital records—Ohio Department of Health death recordsResidents identified in the data source with opioid overdose deaths (OOD)All census OOD included in analysisOhioYes1 January 2010–31 December 2020Young et al. (2021) [32]Retrospective observationalMedical records—EDInjured patients with blood alcohol concentration and urine toxicology tests admitted in 11 American College of Surgeons Level I and II trauma centers across 7 counties in study region 20,448 patients (total); 7707 (Control group); 6022 (Pre-pandemic group), 6719 (Post-pandemic group)Southern CaliforniaYes19 March 2019–30 June 2019 (Historical control); 1 January 2020–18 March 2020 (Pre-pandemic); 19 March 2020–30 June 2020 (Post-pandemic) Zubiago et al. (2021) [47]Retrospective observationalMedical records—EHRPWUD that were hospitalized at TuftsMC with positive toxicology screen for fentanyl, amphetamines, cocaine, opiates, oxycodone, methadone, buprenorphine, benzodiazepines, or alcohol; or having a score of 8 or more on the CIWA scale and/or a score of 2 or more on the CAGE scale; or having been prescribed methadone, buprenorphine, naltrexone, acamprosate, or disulfiram 6637 hospitalizations (Pre-pandemic); 1489 hospitalizations (Post-pandemic)Boston, MassachusettsYes1 January 2017–31 December 2019 (Pre-pandemic); 1 January 2020–31 August 2020 (Post-pandemic)
ijerph-19-08883-t003_Table 3Table 3Data availability by domains.Author (Year)Changes In Substance Use FrequencyChanges in Drug Use Contexts and BehaviorsChanges in Illicit Drug SuppliesChanges in Substance Use Treatment and Harm Reduction Services AccessChanges in Health OutcomesPopulation with History of Substance UseGeneral PopulationAmram et al. (2021) [36]



Yes
Bandara et al. (2020) [42]



Yes
Brothers et al. (2021) [39]



YesYesBuchheit et al. (2021) [61]



Yes
Caton et al. (2021) [38]



Yes
Diaz-Martinez et al. (2021) [29]
Yes
Yes

Downs et al. (2021) [43]



Yes
Duncan et al. (2020) [31]
Yes


YesFrench et al. (2021) [62]



Yes
Glenn et al. 2021 [46]



Yes
Grunvald et al. (2021) [54]




YesHandberry et al. (2021) [49]




YesHerring et al. (2020) [41]



Yes
Holland et al. (2021) [48]




YesHughes et al. (2021) [40]



Yes
Jacka et al. (2021) [25]Yes

YesYes
Janulis et al. (2021) [27]
Yes



Jones et al. (a) (2021) [37]



Yes
Jones et al. (b) (2021) [45]



Yes
Khoury et al. (2021) [55]




YesLucero et al. (2020) [52]




YesMason et al. (2021) [60]




YesMistler et al. (2021) [26]Yes
Yes
Yes
Nguyen et al. (2020) [44]



Yes
Niles et al. (2021) [33]
YesYes


Palamar and Acosta (2020) [30]
Yes
Yes

Palamar et al. (2021) [34]


Yes

Pines et al. (2021) [50]




YesRidout et al. (2021) [58]




YesRosenbaum et al. (2021) [56]




YesShreffler et al. (2021) [53]




YesSoares et al. (2021) [51]




YesStarks et al. (2020) [28]
YesYes


Stroever et al. (2021) [57]




YesVieson et al. (2021) [59]




YesYoung et al. (2021) [32]
Yes



Zubiago et al. (2021) [47]



Yes



## 4. Discussion

Our study provides a characterization of evidence regarding the impacts of the COVID-19 pandemic on a comprehensive range of outcomes concerning illicit substance use within US-specific contexts over the first two waves. Other scoping reviews that shared similar goals have either presented generalized global findings [16] or were narrowed in scope to specific outcomes or study populations (i.e., opioid use-related disorders only) [14,15,17]. Our findings address more discrete research questions to rigorously characterize and complement the evidence (through full data extraction provided in Appendix A–C) and guide subsequent research.

Prevailing evidence presented in this review poses challenges in evaluating the exact magnitude or direction of changes in illicit substance use frequency and behaviors. Except for two studies utilizing biological samples to determine substance use before and post-pandemic [32,33], all studies identified under this domain comprised self-report measures of substance use changes. Results were heterogeneous depending on population subgroups and drug types, although most participants typically indicated no change in use frequency compared to pre-pandemic. Meanwhile, the evidence body for changes in illicit substance use contexts and risk behaviors is limited. Two out of three studies reviewed under this domain pointed to increased harmful substance use behaviors, one with increased uptake of risky drug combinations and fentanyl at the national level [33], and one with increases in self-reported high-risk substance use-associated sexual behaviors among men of sexual minorities [28], whereas the remaining study found no significant changes in drug equipment sharing [26]. Other crucial behavioral outcomes related to illicit drug use, such as changes in other injection-related risks, changes in substance use networks, or changes in use contexts (e.g., using substances alone during isolation) that may increase risks of bloodborne infections such as HCV and HIV or overdose outcomes, were underrepresented in the literature. Finally, with four studies providing data on changes in illicit drug supplies [25,29,30,34], we have an incomplete representation of how pandemic conditions influenced their dynamics within the US. Most notably, empirical data from drug seizures saw an increase in the methamphetamine supply coupled with an increase in fentanyl seizures [34], whereas the other three studies with self-reported viewpoints provided mixed reports regarding the price and quality of the substances of choice [25,29,30].

Although evidence on changes in health services access for those with illicit substance use are more robust compared with the previous domains, the outcomes reported are, once again, varied and fragmented across different study settings. In general, the reduction of limits on MOUD prescribing practices by SAMHSA gave rise to enhanced and lower-barrier access primarily within individual MOUD clinics in terms of buprenorphine initiation, prescription, and dosage duration [31,36,37,38,39,40]. Broader state and nationwide data relating to other aspects of the treatment cascade, including referrals, prescription fillings, and general program capacities, indicate that the regulatory changes were not sufficient to mitigate disruptions of service access relating to the pandemic [41,42,43,44]. Similar patterns could be seen when examining other harm reduction services’ access, albeit with a narrower body of evidence concerning naloxone distribution or syringe services [25,46]. Overall, the evidence suggests relevant and timely policy changes, such as the loosening of buprenorphine regulations, can facilitate greater access to, and retention in, treatment among PWUD, but implementation science studies will be needed to conclusively inform their successful adaptation.

The strongest body of evidence found in this review pertains to changes in substance use-related health outcomes, specifically for overdose outcomes. Among the 10 studies investigating changes in EMS utilization for substance use, ED/EMS overdose data were presented differently, with some focusing on changes in terms of the absolute number and others focusing on the relative proportion of substance use-related encounters. Despite our efforts to reconcile the information through calculating common measures, transparent interpretation proves challenging. Whereas most of the studies showed increasing trends in relative terms, only a subset found absolute increases in the encounters related to substance use [48,53,54,55]. These observations were due, in large part, to a general decrease in ED volumes following the pandemic and thus requires a nuanced understanding of each datapoint within their study contexts. More distinctly, four [53,54,59,60] of the five studies with data on fatal overdoses showed increasing trends associated with the pandemic onset. These findings align with releases of national overdose deaths data showing a record of over 96,000 deaths in 2020 [63] and over 100,000 deaths in the 12 months up to May 2021 [64]. Such increases in negative overdose outcomes nationwide can potentially be explained when they are considered in conjunction with the aforementioned findings concerning potential increases in risky substance use behaviors, increases in the supply of fentanyl, and evidence of reduced harm reduction service capacities due to pandemic disruptions. The contribution of each factor cannot be disentangled through this review and key structural factors not examined here including housing, unemployment, structural racism, and incarceration also likely played a part in the steep rise in mortality [65,66,67,68]. Finally, although data on overdose trends were clear, data on other substance use-related outcomes, such as methamphetamine-related harms (e.g., psychosis episodes), HIV, and HCV outcomes, were not reported in the reviewed studies, as these outcomes are inevitably harder to obtain in comparison to overdose data.

The search strategy and scope of the review in the present study relies on PubMed through LitCovid, which omits any articles that are not indexed in PubMed or other adjacent literature databases. Due to the inclusion criteria specified, the current study also potentially omits any articles presenting evidence on relevant changes in substance use-related outcomes during the time period of interest but without any explicit reference to COVID-19. Likewise, it would be challenging to determine the influence of COVID-19 on the findings of studies not explicitly carried out within the perspective of the pandemic.

In terms of the data presented, we noted the schism of data quality between studies with self-reported measures and process-based measures such as ED/EMS time series data, drug tests results, and drug seizures data, with potential issues of self-reporting bias in the case of the former that can undermine the validity of findings from the respective studies. Similarly, differing interpretations of ED/EMS data across studies in relative compared to absolute terms, as well as data coding inconsistencies between and within jurisdictions, pose challenges in results comparisons. Methodological uncertainties hinder our ability to compare estimates of pandemic-related impacts on outcomes, using pre- and post-pandemic data, due to the diversity of statistical methods employed and variations in the timing and nature of the social distancing measures that were implemented. Importantly, even though we endeavored to harmonize findings by presenting studies that used similar indicators in infographics, there were high levels of heterogeneity in the approaches used to measure substance use outcomes across studies which made comparisons difficult. These graphical representations allowed us to identify trends that should be further examined and provide valuable information to support future meta-analyses using compatible measurements. Most promisingly, quantifiable relative changes in usage between drug types, changes in access measures for harm reduction services (MOUD, syringe services programs, naloxone, etc.), and analyses of non-fatal overdoses as reflective of ED/ER usage changes due to pandemic behavioural responses would be candidate meta-analyses outcomes for future work.

As seen with most other non-cohort studies, ascertaining timely and reliable data on various dimensions of substance use can be exceptionally difficult, thus explaining the predominant use of medical records or cross-sectional survey data as shown by our review. Although such data can provide a good basis for decision-making during urgent times, an over-reliance on healthcare system data may hinder the collection of other important substance use-related outcomes. In addition, our ability to make inferences about the precise impacts of the COVID-19 pandemic on aspects relating to substance use are often limited by the reliance on a pre/post change (also known as ‘before and after’) study design. As a non-randomized approach, this study design carries the risk of bias as a result of confounding factors that distort the comparability of data collected during the different time periods. For this reason, uncontrolled pre/post change studies are heavily discouraged [69]. However, there has been little guidance offered from the general literature on the statistical methods of best practice to mitigate the potential for bias. Consequently, research is needed in this regard to allow quality assessments to be conducted.

We characterized different data sources, study designs, and measures used to investigate the impact of COVID-19 on illicit substance use in the US and identified key outcomes that could be further investigated through systematic reviews/meta-analyses. Future research should focus on utilizing longitudinal study designs from pre-existing cohorts and on developing routine health data collection systems (e.g., in collaboration with harm reduction service providers) to draw more robust inferences on the magnitude and duration of trends in substance use and enable a more effective public health response during crises.

## 5. Conclusions

Since the onset of the COVID-19 pandemic, there have been transformative changes in substance use health outcome trends in the United States. Despite these changes, this review finds limited evidence to demonstrate any corresponding changes in frequency, behaviors, and contexts of illicit substance use. The findings reveal a need for improved data collection practices to facilitate the conduct of timely research and formulation of policy responses to mitigate and prevent the health harms associated with evolving substance use contexts.

## Figures and Tables

**Figure 1 ijerph-19-08883-f001:**
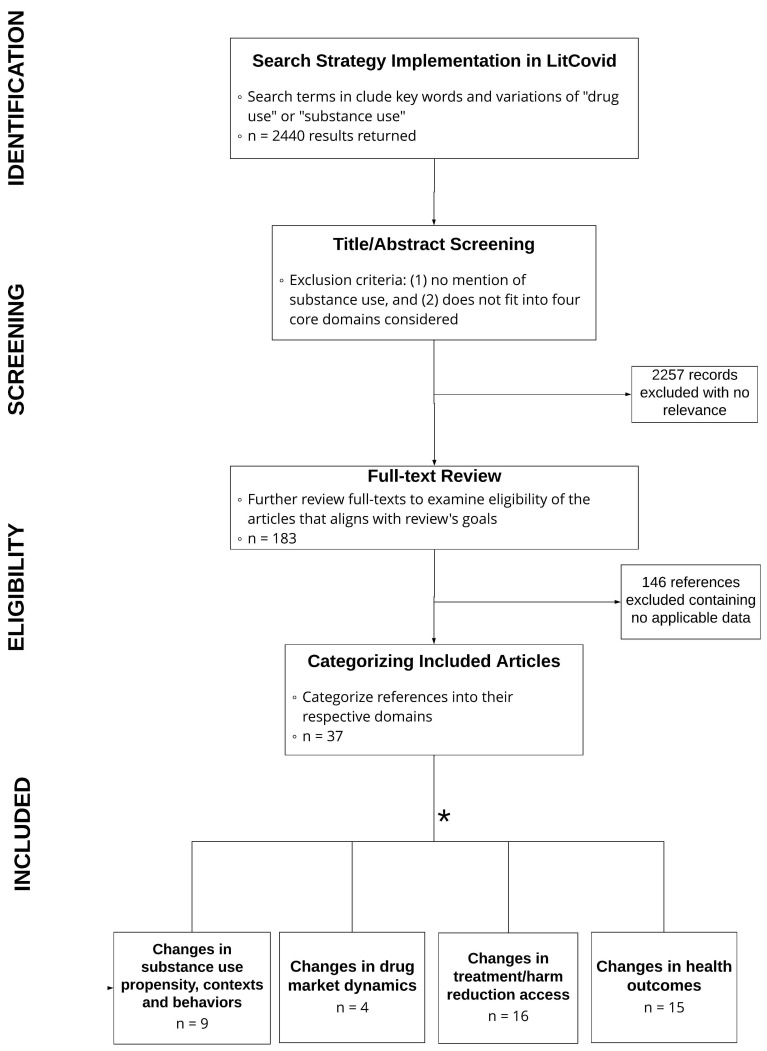
PRISMA literature review process. * Final categories for included articles are not mutually exclusive and articles may be classified in one or more sections.

**Figure 2 ijerph-19-08883-f002:**
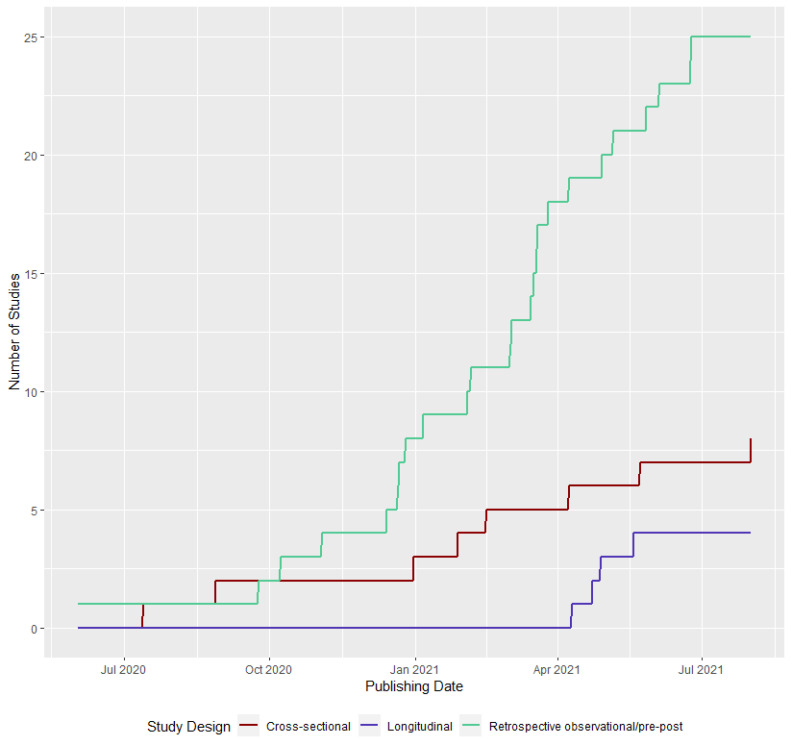
Study designs of selected articles by publishing date.

**Figure 6 ijerph-19-08883-f006:**
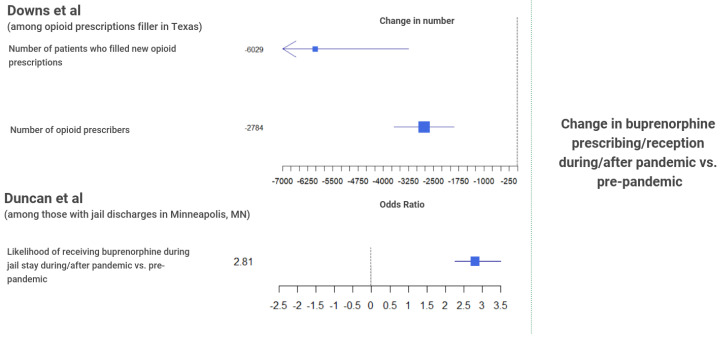
Summary evidence for changes in substance use treatment and harm reduction services access. MN = Minnesota [31,43].

**Table 1 ijerph-19-08883-t001:** Summary of study characteristics included in review (n = 37).

Study Characteristics	N (%)
**Study design**	
Cross-sectional	8 (22%)
Prospective longitudinal	4 (11%)
Retrospective pre-post, using previously collected data	25 (68%)
**Data source**	
Medical records	12 (32%)
Survey data	7 (29%)
Census/surveillance data	5 (14%)
Other	13 (35%)
**US region**	
Midwest	5 (13%)
Northeast	10 (27%)
South	4 (10%)
West	6 (16%)
Overall US	12 (32%)
**Pre- and post-pandemic data availability**	
Yes	29 (78%)

## Data Availability

The dataset used and analyzed for this study can be found under Appendix A.

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
