# Peer review of "Illicit Substance Use and the COVID-19 Pandemic in the United States: A Scoping Review and Characterization of Research Evidence in Unprecedented Times"

_ijerph, 2022, doi:10.3390/ijerph19148883_

Round 1

Reviewer 1 Report

The article is well written, only a few things need to be redone, namely the abstract should not contain the specific markers "Background", "Material" etc. Also, all abbreviated words, please be written on the first use in the text, unfolded, and only then use the abbreviation. This recommendation is for the whole document, not just in the sentences I highlighted. One last aspect, reference 9 is written in round brackets.

Very few mistakes, mostly those mistakes refer to the tamplate of the journal.

See attachment.

Reviewer 2 Report

The paper summarizes the available literature data on Illicit substances used in the first two waves of the COVID-19 pandemic in the United States. The data provided is of real interest and importance in the field. The manuscript is well organized, an adequate approach has been carried out, and the data has been correctly presented and discussed by the Authors.

1.          In the last two years more than 20 review articles have been published on this theme, can you explain how this work stands out from the others.

2.          Why did the authors decide to include only studies from the USA, are the data of the rest of the world irrelevant?

3.          Please can you review figure 1.

“categorize references into their respective domain” n=37

“changes in substances…. N=9; “changes in drug… n=4; “changes in healthcare..” n=16; “changes in health outcomes” n=15

Total= 44

4.          Pleases provide the quality and susceptibility to the bias of the included studies. 

Reviewer 3 Report

Thank you for the opportunity to review this manuscript. This is a scoping review focused in quantitative empirical data regarding the pre-post use of substances in the first two waves of the COVID-19 pandemic in the United States.

During the lockdown measures put in place by governments during this pandemic, important changes in the individuals’ habits were identified, including substance use. A number of studies have assessed the extent to which the frequency of tobacco, alcohol and other drug use increased, and what were the rates of progression to problematic and pathological use. This manuscript provides a global overview of all the empirical studies published to date, using a scoping methodology. Scoping studies provide a preliminary assessment of size and scope of research literature.

This manuscript has different strong points. Overall, the study is clearly conducted. The abstract clearly and accurately describes the contents, a complete description of the protocol used for the review is provided, the methods and results are well written, and the interpretations-conclusions are justified by the results.

Congratulations for the excellent paper. Very interesting and well thought out. I have a few to criticize. I have only some minor points that I consider could make a stronger scientific work:

- Since scoping reviews are not frequently used, I suggest describing the guidelines used for the procedure (concretely, the Munn et al. guidance).

- Figure 1 suggest that studies were only searched into LitCovid. I understand that also Pubmed was searched. Please, clarify.

- Authors did not conduct a formal risk of bias. This is a limitation, and it has not been adequately justified. Please, provide the reasons for this lack.
